# Loneliness is associated with unstable and distorted emotion transition predictions
Ava Q. Ma de Sousa [1] ✉, Miriam E. Schwyck [2], Laura Furtado Fernandes [3], Ezra Ford [3], Begüm G. Babür [4], Chang Lu[4], Jacob C. Zimmerman [4], Hongbo Yu [1], Shannon M. Burns [3,5] ✉ & Elisa C. Baek [4] ✉

Loneliness is associated with disruptions in socio-cognitive processes, including altered self-other representations and atypical processing of external stimuli. Here, we examine whether loneliness is characterized by altered expectations of emotion transitions for both oneself and others, which may contribute to the observed disruptions in socio-cognitive processes and pose challenges for social connection. Drawing on data from seven studies (total $N = 1730$; $N_{Study1} = 113$; $N_{Study2} = 185$; $N_{Study3} = 376$; $N_{Study4} = 91$; $N_{Study5} = 68$; $N_{Study6} = 41$; $N_{Study7} = 856$) using a validated emotion transition task, we found that lonely individuals hold atypical expectations about both their own and others' likelihoods to transition between emotions and are less accurate at predicting others' emotion transitions. While lonely participants relied less on their own emotion transition patterns when predicting others' emotions, they also showed a response pattern that may reflect reduced confidence, suggesting they use a less stable or altered strategy for predicting others. Furthermore, lonely individuals perceived others as more volatile, expecting them to switch emotion valence states more frequently and be less likely to maintain the same emotion state. At the same time, they viewed themselves as more likely to shift away from positive states. Altogether, these findings suggest that loneliness is associated with unstable, inaccurate expectations of emotion continuity in others and a bias against sustaining positive emotions in the self—patterns that may contribute to challenges in social interactions and reinforce feelings of isolation.

The detrimental consequences of loneliness are well-established; loneliness has been linked to negative outcomes for mental and physical health, including cardiovascular disease[1] and premature mortality[2]. These negative effects of loneliness are particularly concerning given its widespread prevalence[3,4]. Thus, gaining a deeper understanding of the cognitive mechanisms of loneliness is crucial, as it can provide insights into how feelings of social disconnection persist and guide efforts for interventions to improve well-being.

At its core, loneliness is a subjective experience of social disconnection, characterized by a sense of not being understood by others[5]. Successful social interactions, however, rely on the ability to both understand and be understood by others[6], which requires not only accurate representations of

self and others but also the ability to track and anticipate changes in a social partner's emotions and behaviors. Predicting emotion transitions, such as whether someone will become angry or remain calm, enhances social decision-making by facilitating spontaneous adaptation to interaction partners' emotions and behaviors[7]. This fosters interpersonal connection[8,9] and promotes positive social outcomes, such as increased affiliation and smooth social interactions[10–12]. Thus, one reason for persistent feelings of social disconnection in loneliness may be due to disruptions in emotion prediction that impair social interactions and reinforce a chronic sense of not being understood.

Emerging evidence supports this theory: people typically infer others' emotions using external cues and internal models based on personal

[1]Department of Psychological and Brain Sciences, University of California, Santa Barbara, Santa Barbara, CA, USA. [2]Department of Psychology, Columbia University, New York, NY, USA. [3]Department of Psychological Science, Pomona College, Pomona, CA, USA. [4]Department of Psychology, University of Southern California, Los Angeles, CA, USA. [5]Department of Neuroscience, Pomona College, Pomona, CA, USA. ✉e-mail: avamadesousa@ucsb.edu; shannon.burns@pomona.edu; elisa.baek@usc.edu

experience[13–15]. However, loneliness is linked with idiosyncratic mental processing of external stimuli[16], maladaptive emotion processing including heightened sensitivity to social threats[17,18], and increased activation in the brain's negative-emotion circuitry[19]. These altered cognitive processes suggest that relying on one's own internal affective models to predict others' emotions is unlikely to be an adaptive strategy for lonely individuals. Indeed, prior work suggests that loneliness is associated with reduced accuracy in forecasting emotion transitions of others[13]. However, the mechanisms behind emotion prediction disruptions in loneliness remain unclear. One possibility is that lonely individuals rely on their own idiosyncratic internal models to predict others' emotions, leading to systematic inaccuracies. Alternatively, they may recognize the discrepancy between their own experiences and those of others, resulting in a reduced reliance on self-referential models and suggesting the involvement of different predictive strategies. In the present investigation, we seek to characterize the predictions that lonely individuals make about their own and others' emotions. To do so, we draw on data from seven studies utilizing a well-validated emotion transition task, in which participants estimate the likelihood that a target – either themselves or a generic, typical other – will transition between different emotion states.

First, we examine whether lonely individuals show atypical patterns in estimating emotion transitions, reflecting their expectations of how emotions change for both themselves and others. While loneliness has been linked to atypical processing in other domains[16,20], it remains unclear whether this atypicality extends to emotion transitions specifically. Next, we test whether lonely individuals' subjective expectations of emotion transitions are less aligned with *actual* emotion transitions of others – in other words, whether they demonstrate lower accuracy. We then investigate whether inaccuracies arise from lonely individuals relying on their own atypical emotion models to predict other people's emotional states. Next, we test whether atypical emotion transition expectations and reduced accuracy are driven by differences in how lonely individuals perceive transition valence patterns in the self and others.

## Methods

### Participants

We include data from seven studies comprising a total of 1730 participants. Data collection for the studies occurred between 2018 and 2024. Sample sizes for each study were as follows: Study 1 ($N = 113$), Study 2 ($N = 185$), Study 3 ($N = 376$), Study 4 ($N = 91$), Study 5 ($N = 68$), Study 6 ($N = 41$) and Study 7 ($N = 856$) (see Table 1). Participants were recruited via Amazon Mechanical Turk (Studies 1 and 2), university student pools (Studies 3, 4, 5 and 6) and Prolific (Study 7). Inclusion criteria required participants to pass attention checks and complete at least 50% of the study. See Table 2 for self-reported demographic information for the participants in each study. All studies were approved by the corresponding Institutional Review Board at the location of data collection: Princeton University (Studies 1−4), University of Southern California (Study 5), and Pomona College (Studies 6 and 7). Data from Studies 1−4 were publicly available at [13,21] and data from Studies 5−7 were collected by the authors. Participants provided informed consent in accordance with procedures approved by the relevant IRB. Participants were compensated monetarily or with course credit. All data and code used in this investigation can be accessed on the Open Science Framework (data from Study 5: https://osf.io/7gjz9/; data from Studies 6 and 7: https://osf.io/7ybwr/; analysis code: https://osf.io/7gjz9/).

### Materials

**Emotion transition task**. All participants completed a version of the emotion transition task[22]. In this task, participants rate the likelihood that a target would transition from one emotion to another. The emotion transition task has been established as a reliable and externally valid method to measure individual differences in emotion prediction, as its ratings align with actual emotion transitions observed through Ecological Momentary Assessment (EMA) methods[22]. In every version of the task, participants made ratings for themselves and another person. In Studies 1, 2, 3, 4, and 6,

**Table 1 | Study characteristics**

| Study | N | Participant Source | Emotions Included | Type of Other | $M_{Loneliness}$ (SD) | Cronbach's $\alpha_{Loneliness}$ |
|---|---|---|---|---|---|---|
| Study 1 | 113 | MTurk | anxious, calm, happy, irritable, sad, sluggish, full of thought | Average generic other | 19.4 (13.76) | 0.947 |
| Study 2 | 185 | MTurk | anxious, calm, happy, irritable, sad, sluggish, full of thought | Average generic other | 23.23 (13.53) | 0.937 |
| Study 3 | 376 | Princeton undergraduates | anxious, calm, happy, irritable, sad, sluggish, full of thought | Average generic other | 19.62 (9.88) | 0.916 |
| Study 4 | 91 | Princeton undergraduates | assertive, confident, grouchy, sad, unrestrained, bold, irritable, lively, nervous, talkative, contempt, disgust, embarrassment, love, satisfaction | Average generic other | 23.01(9.83) | 0.933 |
| Study 5 | 68 | USC undergraduates | anxious, calm, happy, irritable, sad, sluggish, alert | Average community member (undergraduate) | 16.97 (11.45) | 0.943 |
| Study 6 | 41 | Pomona College undergraduates | anxious, sad, amused, satisfied, hopeful, content | Generic other | 19.27 (10.54) | 0.938 |
| Study 7 | 856 | Prolific | anxious, sad, amused, satisfied, hopeful, content | Previously seen specific other | 24.51 (12.50) | 0.928 |

Note. $M_{Loneliness}$ mean loneliness score, SD standard deviation.

**Table 2 | Demographic Characteristics**

| Sample characteristics | Study | | | | | | |
|---|---|---|---|---|---|---|---|
| | **1** | **2** | **3** | **4** | **5** | **6** | **7** |
| N | 113 | 185 | 376 | 91 | 68 | 41 | 856 |
| $M_{age}$ (SD) | 37.24 (10.19) | 35.81 (10.66) | 19.43 (1.23) | 18.56 (2.08) | 18.82 (0.98) | 19.15 (1.00) | 36.03 (12.52) |
| Sex/Gender | | | | | | | |
| Female/Woman | 49 (43.4%) | 67 (36.2%) | 253 (67.3%) | 54 (59.3%) | 18 (26.5%) | 29 (70.7%) | 560 (65.3%) |
| Male/Man | 61 (54.0%) | 114 (61.6%) | 123 (32.7%) | 36 (39.6%) | 49 (72.1%) | 9 (22.0%) | 257 (30.0%) |
| Other | 1 (0.9%) | 4 (2.2%) | 0 (0%) | 1 (1.1%) | 1 (1.5%) | 3 (7.3%) | 35 (4.2%) |
| Not stated | 2 (1.8%) | 0 (0%) | 0 (0%) | 0 (0%) | 0 (0%) | 0 (0%) | 4 (0.4%) |
| Ethnicity | | | | | | | |
| Hispanic/Latinx | 12 (10.6%) | 19 (10.3%) | 45 (12.0%) | 20 (22.0%) | 21 (30.9%) | 4 (9.8%) | 74 (8.6%) |
| Not Hispanic/Latinx | 99 (87.6%) | 163 (88.1%) | 318 (84.6%) | 71 (78.0%) | 47 (69.1%) | 37 (90.2%) | 774 (90.4%) |
| Not stated | 2 (1.8%) | 3 (1.6%) | 13 (3.5%) | 0 (0%) | 0 (0%) | 0 (0%) | 8 (0.9%) |
| Race | | | | | | | |
| American Indian/Alaska Native | 2 (1.8%) | 1 (0.5%) | 3 (0.8%) | 0 (0%) | 1 (1.5%) | 0 (0%) | 8 (0.9%) |
| Asian | 7 (6.2%) | 10 (5.4%) | 124 (33.0%) | 37 (40.7%) | 30 (44.1%) | 12 (29.3%) | 82 (9.5%) |
| Black/African American | 12 (10.6%) | 21 (11.4%) | 22 (5.8%) | 3 (3.3%) | 7 (10.3%) | 2 (4.9%) | 163 (19.0%) |
| Native Hawaiian/Pacific Islander | 0 (0%) | 1 (0.5%) | 1 (0.3%) | 1 (1.1%) | 0 (0%) | 0 (0%) | 1 (0.1%) |
| White | 89 (78.8%) | 142 (76.8%) | 185 (49.2%) | 39 (42.9%) | 13 (19.1%) | 16 (39.0%) | 481 (56.2%) |
| Other | 0 (0%) | 10 (5.4%) | 36 (9.6%) | 11 (12.1%) | 16 (23.5%) | 11 (26.8%) | 112 (13.1%) |
| Not stated | 3 (2.7%) | 0 (0%) | 5 (1.3%) | 0 (0%) | 0 (0%) | 0 (0%) | 9 (1.1%) |

the other target was an "average other person", and in Study 5 the target was a typical member of the participants' community (in this case, a typical undergraduate student at the institution). In Study 7, the target was a specific stranger that participants had learned about earlier in the study. Given that this variation of the "other" target deviates meaningfully from that of Studies 1−6 by asking about a specific fictional individual, we did not include the "other" ratings from Study 7 in any of our analyses.

On each trial, participants were presented with two mental states: the first representing the target person's current state and the second a potential future state. They then rated the likelihood that the target would transition between these states on a continuous scale from 0%−100%. The number and type of transitions varied across studies. In Studies 1, 2, 3, and 5, participants rated transitions between all possible pairs of seven emotion states, including the likelihood of remaining in the same state. In Study 4, participants rated a total of 75 transitions which were drawn from a subset of possible transitions among 15 different emotions. In Studies 6 and 7, participants rated only transitions between six different emotion states (i.e., excluding same-state transitions) resulting in 30 total transitions. Although the specific mental states varied across studies, all sets included positive (e.g., "happy"), negative (e.g., "irritable"), and neutral states (e.g., "full of thought") (see Table 1). In every study, the order in which transitions were presented was randomized for each participant.

**Loneliness.** In all studies, loneliness was measured using the 20-item UCLA Loneliness scale[23]. This scale assesses various dimensions of loneliness by asking participants to indicate how frequently they experience each described statement (e.g., "How often do you feel that you are 'in tune' with the people around you?"). Responses were measured on a Likert scale, with "0 = Never" to "3 = Often" (see Table 1 for mean and SD of loneliness scores).

**Data analyses.** All analyses were performed using R (version 4.3.1). Visualizations were created with the jtools[24] and ggplot2[25] packages. Detailed descriptions for each set of analyses follow below. Analyses were not preregistered.

**Loneliness and typicality**

We first tested whether lonely individuals tend to experience and expect emotion transitions that are atypical compared to non-lonely individuals, as indicated by less typical emotion transition ratings for self and others. To quantify self typicality, we first computed the group average vector of all self-targeted emotion transition ratings for each study. We then correlated each participant's vector of self-targeted ratings with the group average to compute a typicality index per participant. High scores on this index indicate that participants' self emotion transition likelihood ratings were very similar to how other people rated their own emotion transitions on average. We then correlated this typicality index with participants' standardized loneliness scores. We also did this procedure for other-targeted ratings, which allowed us to test the relationships between loneliness and typicality in personal emotion transition experience and between loneliness and typicality in expectations of others. We also conducted Intersubject Representational Similarity Analysis (IS-RSA) to further investigate the pattern of atypicality, specifically to determine whether lonely individuals hold unique, idiosyncratic expectations of emotion transitions. Details of this analysis are provided in the Supplementary Materials.

**Loneliness and accuracy of others' emotions**

To investigate whether loneliness was linked with less accurate predictions of others' emotion transition likelihoods, we calculated a "ground truth" likelihood value for each unique emotion transition pair that represents the average self-rating across all participants in a study for that transition. We then fit a linear mixed effects model for each study using the lme4 package[26] predicting the "ground truth" likelihood of each unique transition from participants' other-targeted ratings of that transition, participants' loneliness scores, and the interaction of these terms. Random intercepts were included in the model for participant, emotion state transitioned from, and emotion state transitioned to in order to account for nonindependence in our data due to repeated observations for each participant and emotion. A significant interaction term here indicates that loneliness moderates the strength of the relationship between predicted and actual emotion transition likelihoods for others (i.e., accuracy of prediction).

## Loneliness and anchoring on one's own emotion transitions

We next tested whether loneliness moderates the degree to which individuals use their own emotion transition likelihoods to predict others' emotion transition likelihoods. Similar to our analysis assessing the relationship between loneliness and accuracy, for each study, we fit a linear mixed effects model predicting each participant's other-targeted emotion ratings from self-targeted emotion ratings, loneliness, and their interaction, with random intercepts for participant, emotion item transitioned from, and emotion item transitioned to.

## Loneliness and perceived volatility of others' emotions

To investigate the role of valence in lonely individuals' predictions of others' emotions, we first categorized each emotion in the emotion transition tasks as positive, negative, or neutral (see Supplementary Table 1). Given that we were interested in shifts across valence types, we excluded neutral emotions from our analyses. We then classified each emotion transition into one of four types: positive-to-positive (e.g., calm to happy), positive-to-negative (e.g., calm to irritable), negative-to-negative (e.g., irritable to anxious), and negative-to-positive (e.g., anxious to calm). We then fit a linear mixed effects model for each study predicting participants' ratings of emotion transitions for others from the categorical variable for emotion transition type, loneliness, and their interaction with random intercepts for participant, emotion item transitioned from, and emotion item transitioned to. Simple slopes were calculated for each transition category with the reghelper package[27].

## Loneliness and confidence of one's own ratings

To examine whether lonely individuals were less confident in their own and others' emotion transitions, we operationalized confidence as the standard deviation of transition likelihood estimations for each participant. We inferred that a more confident participant would assign high likelihood ratings to some emotion transition pairs and low likelihood ratings to others, whereas a less confident participant would provide less variable ratings overall. We did this separately for self and other-targeted ratings, generating one value for each participant that represented their confidence for self emotion transitions and one value that represented their confidence for others' emotion transitions. These participant-level confidence measures were then correlated with loneliness separately for each study.

## Loneliness and valence of self emotions

Finally, we examined the role of valence in lonely individuals' own emotion transitions. Analogous to our approach investigating the relationship between loneliness and the valence of others' emotion transitions, for each study, we fit a linear mixed effects model predicting participants' self-targeted ratings of emotion transitions from the categorical variable for emotion transition type, loneliness, and their interaction with random intercepts for participant, emotion item transitioned from, and emotion item transitioned to. Simple slopes were calculated for each transition category with the reghelper package[27].

## Meta-analytic approach

Having obtained study-specific estimates for each question of interest as described above, we then estimated the overall effect sizes across studies using random-effects meta-analytic models for each set of analyses. Specifically, we used restricted maximum likelihood (REML) estimation with the metafor package[28] to model the between-study variability. This approach yielded pooled estimates with 95% confidence intervals reflecting the overall interest effects while accounting for heterogeneity across studies. For all analyses involving self-transition ratings, data from all seven studies were included ($k = 7$). For all analyses involving other-transition ratings, data from six studies were included as Study 7 did not collect comparable data for others' transitions ($k = 6$).

## Statistical assumptions and model diagnostics

We performed the following assumption checks for all analyses conducted on individual datasets: for every Pearson correlation and simple linear regression we applied Shapiro-Wilk tests to the focal variables and Breusch-Pagan tests to the residuals; for each linear mixed-effects model we assessed residual normality with the DHARMa simulated-residual Kolmogorov-Smirnov (KS) test and inspected DHARMa dispersion statistics and residual Breusch-Pagan tests, using the dharma[29] and lmertest[30] packages.

Across the seven datasets a substantial proportion of Shapiro-Wilk and DHARMa KS tests were significant, indicating modest departures from strict normality. In contrast, few Breusch-Pagan or DHARMa dispersion ratios were significant, providing little evidence of heteroscedasticity or over-dispersion (see Supplementary Tables 2–6 for full results). As each study had >40 participants, and diagnostic plots showed acceptable skew or kurtosis, we retained the planned Pearson correlations and Gaussian linear (mixed) models.

The robustness of our inference is further enhanced by the meta-analytic approach we focus on here, as combined effect sizes are based on the aggregate across all studies, downweighing studies whose standard errors may be inflated by modest assumption violations and cushioning the influence of residual non-normality in single studies. In all meta-analyses we estimated the between-study variance $\tau^2$ with restricted maximum likelihood and report both $\tau^2$ and the $I^2$ in Results to describe residual heterogeneity for each pooled effect.

## Results

We focus on results from the internal meta-analytic approach which leverages data from all seven studies to synthesize effect sizes across all studies to obtain an estimate of overall effects; results from models that were fit separately for each study can be found in the Supplementary Information.

## Lonely individuals show atypical expectations of emotion transitions for self and others

We first tested whether lonely individuals tend to experience and expect emotion transitions that are atypical compared to non-lonely individuals, as indicated by less normative emotion transition ratings for self and others. As expected, loneliness was linked to less normative emotion transition expectations for self and others. Compared to their non-lonely counterparts, lonely individuals' expectation of transition likelihood between emotions for self was less similar to the group average ($\beta = -0.131$, $SE = 0.039$, $p = 0.001$, 95% CI [-0.207, -0.055], $\tau^2 = 0.005$, $I^2 = 47.0\%$) (see Fig. 1A, B). Similarly, lonely individuals' ratings for emotion transitions of others also deviated from group norms ($\beta = -0.169$, $SE = 0.034$, $p < 0.001$, 95% CI [-0.237, -0.102], $\tau^2 < 0.001$, $I^2 = 0.00\%$) (see Fig. 1C, D). These results suggest that loneliness is associated with atypical expectations for both one's own and others' emotion transitions. (See Supplementary Fig. 1 for individual study estimates of typicality, Supplementary Fig. 2 for results controlling for age, and Supplementary Fig. 3 for transition-level results).

We also conducted IS-RSA to test whether these results follow an "Anna Karenina" pattern[31], or the notion that non-lonely individuals are all alike, whereas every lonely individual has their own unique, idiosyncratic expectations of emotion transitions. Our results support this hypothesis, suggesting that lonely individuals deviate not only from group norms and from non-lonely individuals, but also from one another. Results supported this conclusion for both self ($\rho = -0.135$, SE = 0.044, $p = 0.002$, 95% CI [-0.221, -0.048], $\tau^2 = 0.007$, $I^2 = 57.83\%$) and other $\rho = -0.163$, SE = 0.033, $p < 0.001$, 95% CI [-0.228, -0.097], $\tau^2 < 0.001$, $I^2 = 0.00\%$ transition ratings. In other words, loneliness was linked to a greater degree of interpersonal variability in emotion transition expectations. See "Anna Karenina Typicality Analysis" in the Supplementary Information and Supplementary Fig. 4 for more details.

## Lonely individuals are less accurate in their predictions of others

Given our findings suggesting that loneliness is linked with atypical expectations of emotion transitions for both self and others, we next asked if lonely individuals were also less accurate in their predictions of others' emotion transition likelihoods. In line with our hypothesis and replicating prior work[12,13], we found a significant interaction of loneliness and participants' ratings of others' emotion transition likelihoods, such that lonely

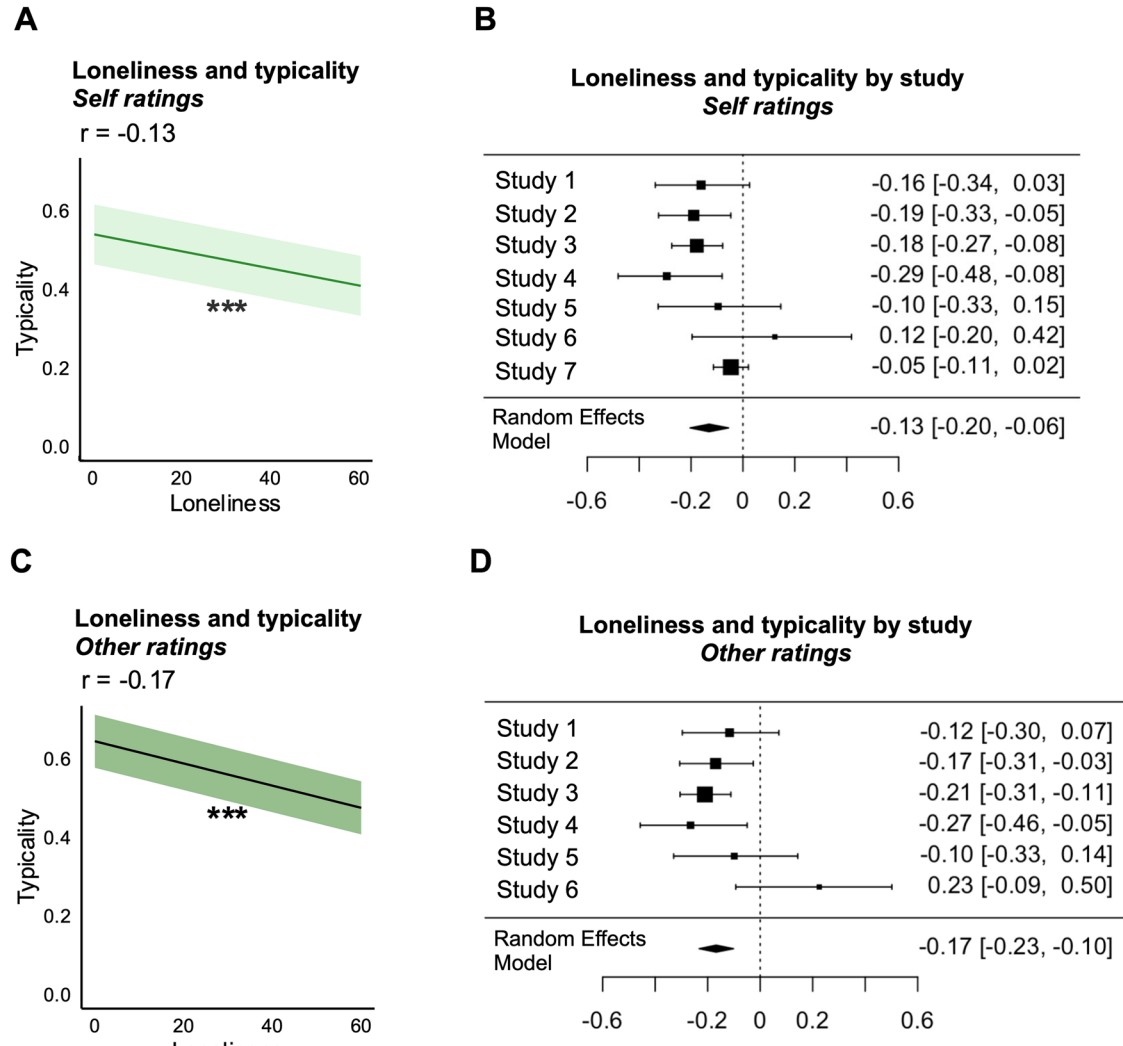

**Fig. 1 | Loneliness is associated with atypicality of emotion transition ratings for self and others. A** Loneliness is negatively associated with typicality of emotion transition ratings for the self. The dark green line represents the meta-analytic correlation estimate, and the light green bands indicate the 95% confidence interval. Higher loneliness was associated with lower typicality of self emotion transition ratings, suggesting that lonely participants' ratings of their own emotions are less normative ($N = 7$ studies). Asterisks indicate significance level: $p < 0.001$ (***). **B** Correlation estimates from individual studies of the relationship between loneliness and typicality of emotion transition ratings for the self are visualized. Each square represents the correlation estimate from the respective dataset with error bars denoting the corresponding 95% confidence intervals ($N_{Study1} = 113$; $N_{Study2} = 185$; $N_{Study3} = 376$; $N_{Study4} = 91$; $N_{Study5} = 68$; $N_{Study6} = 41$; $N_{Study7} = 856$). The rhombus at the bottom represents the overall meta-analytic correlation estimate with its midpoint indicating the average effect size and its width representing the 95%

confidence interval, summarizing the effect size across all included studies. **C** Loneliness was negatively associated with typicality of emotion transition ratings for others. The black line represents the meta-analytic correlation estimate, and the green bands indicate the 95% confidence interval. Higher loneliness was associated with lower typicality ratings for others, suggesting that lonely participants' perceptions of others' emotions are less normative ($N = 6$ studies). Asterisks indicate significance level: $p < 0.001$ (***). **D** Correlation estimates from individual studies of the relationship between loneliness and typicality of emotion transition ratings for others are visualized. Each square represents the correlation estimate from a single dataset with error bars denoting the corresponding 95% confidence intervals ($N_{Study1} = 113$; $N_{Study2} = 185$; $N_{Study3} = 376$; $N_{Study4} = 91$; $N_{Study5} = 68$; $N_{Study6} = 41$). The rhombus at the bottom represents the overall meta-analytic correlation estimate with its midpoint indicating the average effect size and its width representing the 95% confidence interval, summarizing the effect size across all included studies.

individuals' ratings of others were less aligned with the group's average ratings for self ($\beta = -0.016$, $SE = 0.007$, $p = 0.029$, 95% CI [-0.029, -0.002], $\tau^2 < 0.001$, $I^2 = 67.0\%$) (Fig. 2). (See Supplementary Fig. 5 for visualizations of the meta-analytic main effects, Supplementary Fig. 6 for study-specific estimates of the interaction of accuracy and loneliness, Supplementary Table 7 for study specific model results, Supplementary Fig. 7 for results controlling for age, and Supplementary Fig. 8 for transition-level results).

**Lonely individuals anchor less on their own emotion transitions to predict others**

Prior research suggests that using oneself as a reference point when forming expectations for others is generally an effective strategy[32]. However, we

found that lonely individuals exhibit less typical expectations of emotion transitions for themselves, which may limit the effectiveness of using their own experiences as an anchor when predicting others' emotions. Thus, if lonely individuals were to rely on their own emotion transition patterns, they would be less accurate in predicting others' emotion transitions. An alternative possibility is that lonely individuals may adopt a different approach and rely less on their own emotion transitions when predicting others' emotions. To arbitrate between these possibilities, we next tested whether loneliness moderates the degree to which individuals rely on their own emotion transitions to predict those of others. We found that loneliness indeed affects this relationship, such that lonely individuals anchor less on their own emotion transition ratings when predicting others' transitions

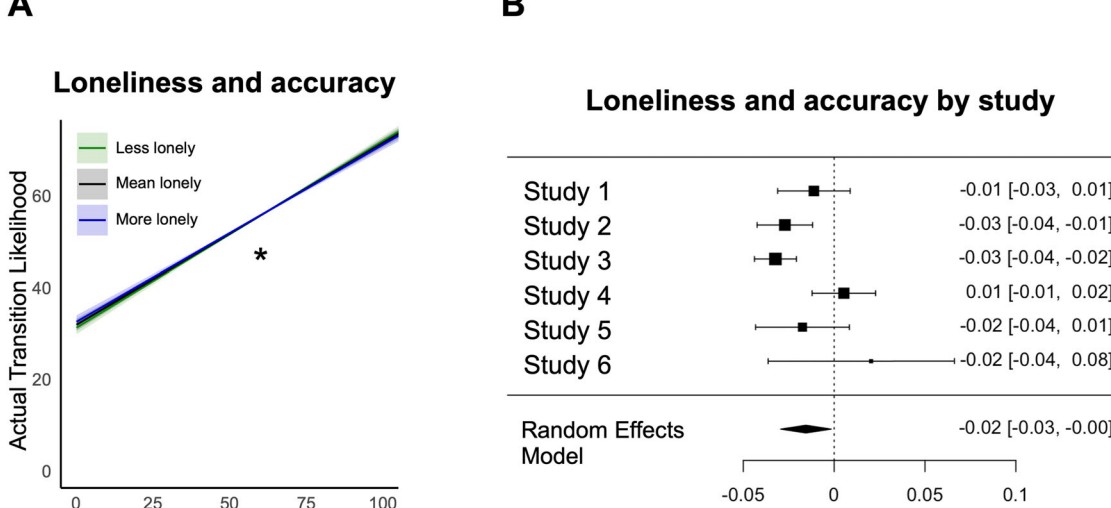

**Fig. 2 | Loneliness is associated with lower accuracy in predicting other's emotion transitions. A** The lines represent the predicted relationship between predicted emotion transition ratings for a typical other and a proxy 'ground truth' for actual emotion transitions of others at different levels of loneliness: lower loneliness in green (-1 SD), average in black (mean), and higher loneliness in blue (+ 1 SD). The shaded areas indicate the 95% confidence intervals (N = 6 studies). This significant interaction suggests that lonelier individuals are less accurate in their predictions. Asterisk indicates significance level: $p < 0.05$ (*). **B** Individual study estimates of the

interaction effect are visualized. Each square represents the estimated interaction effect from a single dataset with error bars indicating the corresponding 95% confidence intervals ($N_{Study1}$ = 113; $N_{Study2}$ = 185; $N_{Study3}$ = 376; $N_{Study4}$ = 91; $N_{Study5}$ = 68; $N_{Study6}$ = 41). The rhombus at the bottom represents the overall meta-analytic estimate with its midpoint indicating the pooled effect size and its width representing the 95% confidence interval, summarizing the effect across all included studies.

($\beta$ = -0.035, SE = 0.007, $p < 0.001$, 95% CI [-0.048, -0.022], $\tau^2 = 0$, $I^2$ = 65.0%) (Fig. 3). Thus, our results suggest that lonely individuals may be aware that their own emotion transitions are less typical and therefore rely less on their own experiences to predict others' emotions. (See Supplementary Fig. 9 for meta-analytic main effects, Supplementary Fig. 10 for study-specific estimates of the interaction of anchoring and loneliness, Supplementary Table 8 for study specific models, and Supplementary Fig. 11 for transition-level results).

**Lonely individuals view others to be volatile and are less confident in their predictions**

Although lonely individuals rely less on their own atypical models when predicting others' emotion transitions, their expectations remain less accurate. Accordingly, we sought to further characterize the mental models lonely people may be using to predict others' emotions. Given that loneliness is associated with increased attention to and recall of negative information[33],[34], we investigated the role of valence in lonely individuals' predictions of emotion transitions. Loneliness was associated with greater expectations that others would transition between valence states, perceiving shifts from positive-to-negative states ($\beta$ = 0.045, SE = 0.021, $p$ = 0.029, 95% CI [0.005, 0.085], $\tau^2$ = 0.000, $I^2$ = 0.00%) and negative-to-positive states ($\beta$ = 0.041, SE = 0.019, $p$ = 0.029, 95% CI [0.004, 0.078], $\tau^2$ = 0.000, $I^2$ = 0.63%) as more likely. Loneliness was also associated with *lower* expectations that others would remain in a positive state (i.e., positive-to-positive) ($\beta$ = -0.108, SE = 0.016, $p < 0.001$, 95% CI [-0.140, -0.076], $\tau^2$ = 0.00, $I^2$ = 0.00%) or a negative state (i.e., negative-to-negative) ($\beta$ = -0.037, SE = 0.014, $p$ = 0.009, 95% CI [-0.064, -0.009], $\tau^2 < 0.001$, $I^2$ = 0.04%). Thus, lonely individuals perceived others as more emotionally volatile, expecting them to transition between valence states more frequently and to be less likely to remain in the same valence state (Fig. 4A, B). (See Supplementary Table 9 for study-specific models and Supplementary Table 10 and Supplementary Fig. 12 for study-specific simple slopes). Using Thornton and Tamir's 3 d Mind Model[3][5],[36], we further quantified the "distance" between emotional states as an additional index of volatility. Consistent with our earlier findings, lonely individuals perceived others as exhibiting larger jumps along the valence

dimension, indicating that they expect more pronounced shifts in others' emotional states (see Supplementary Figs. 13–19).

We next examined individuals' confidence in their emotion transition ratings. Specifically, we wanted to understand whether lonely individuals' volatile mental model emerges because they are less confident about what emotion transitions will happen generally. Indeed, loneliness was negatively associated with confidence in ratings for others' emotion transitions ($\beta$ = -0.114, SE = 0.034, $p < 0.001$, 95% CI [-0.182, -0.047], $\tau^2 < 0.001$, $I^2$ = 0.00%) (see Supplementary Fig. 20; for study-specific visualizations of correlations, see Supplementary Fig. 21). These results suggest that the expectations of volatility displayed by lonely individuals are accompanied by a lack of confidence in their predictions of others' emotion transitions.

**Lonely individuals view themselves as less stable in positive states**

Finally, we examined the role of valence in lonely individuals' emotion transitions for the self. Loneliness was associated with greater expectations of transitioning out of a positive state, with both an increased expectation that the self would transition from a positive state to a negative state ($\beta$ = 0.133, SE = 0.027, $p < 0.001$, 95% CI [0.080, 0.185], $\tau^2$ = 0.003, $I^2$ = 61.93%) and a decreased expectation of staying in a positive state (i.e., positive-to-positive) ($\beta$ = -0.133, SE = 0.023, $p < 0.001$, 95% CI [-0.179, -0.088], $\tau^2$ = 0.003, $I^2$ = 67.6%). Loneliness was also associated with greater likelihood of staying in a negative state (i.e., negative-to-negative), although this relationship was marginally statistically significant ($\beta$ = 0.061, SE = 0.033, $p$ = 0.065, 95% CI [-0.004, 0.126], $\tau^2$ = 0.006, $I^2$ = 0.77%). There was no statistically significant effect of loneliness on negative-to-positive transitions ($\beta$ = -0.009, SE = 0.027, $p$ = 0.737, 95% CI [-0.061, 0.043], $\tau^2$ = 0.003, $I^2$ = 64.5%). Thus, lonely individuals perceived themselves as less likely to remain in positive emotional states, exhibiting both a higher likelihood of transitioning from a positive to a negative state and a lower likelihood of staying in a positive state (Fig. 4C & D). They also exhibited a tendency to see themselves as persisting in negative states. (See Supplementary Tables 11 and 12 and Supplementary Fig. 22 for study-specific results). Our results therefore suggest that loneliness may be characterized by a sense of

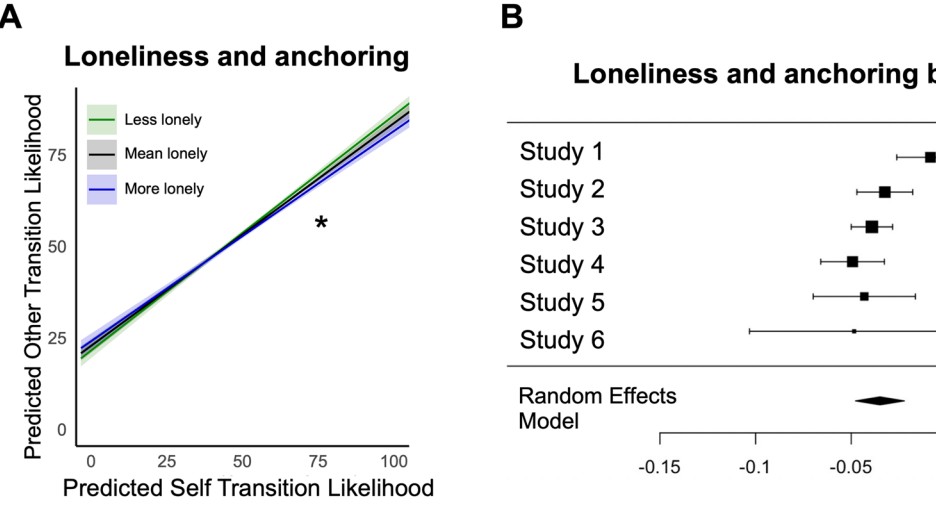

**Fig. 3 | Lonely individuals anchor less on themselves to predict others' emotion transitions. A** The lines represent the predicted relationship between emotion transition ratings for the self and emotion transition ratings for others at different levels of loneliness: lower loneliness in green (-1 SD), average in black (mean), and higher in blue ( + 1 SD). The shaded areas indicate the 95% confidence intervals ($N$ = 6 studies). This significant interaction suggests that loneliness moderates the extent to which individuals anchor their expectations of others' emotion transitions on their own. Asterisk indicates significance level: $p < 0.05$ (*). **B** Individual study

estimates of the interaction effect are visualized. Each square represents the estimated interaction effect from a single dataset with error bars indicating the corresponding 95% confidence intervals ($N_{Study1}$ = 113; $N_{Study2}$ = 185; $N_{Study3}$ = 376; $N_{Study4}$ = 91; $N_{Study5}$ = 68; $N_{Study6}$ = 41). The rhombus at the bottom represents the overall meta-analytic estimate with its midpoint indicating the pooled effect size and its width representing the 95% confidence interval, summarizing the effect across all included studies.

instability in positive emotions, which could contribute to challenges in sustaining feelings of happiness and emotional well-being. Additional analyses in using the 3 d Mind Model[35,36] further support this finding of "anti-positivity" (see Supplementary Figs. 23–29).

We again examined the association between loneliness and confidence in ratings, this time for estimates of the self. Mirroring our results for ratings of others' emotion transitions, we found that loneliness was negatively associated with confidence of ratings for self ($\beta$ = -0.080, $SE$ = 0.024, $p$ = 0.001, 95% CI [-0.127, -0.022], $\tau^2 < 0.001$, $I^2$ = 0.00%) (see Supplementary Fig. 30 and Fig. 31 for study specific correlations). This suggests that lonely individuals' ratings are accompanied by a lack of confidence in their predictions of their own emotion transitions.

## Discussion

How do lonely individuals process their own and others' likelihood to transition between emotions, and what might this reveal about the experience of loneliness itself? Here, we investigated whether loneliness is associated with altered expectations of how emotions change over time in both the self and others. Using an internal meta-analysis approach[37] across seven studies, we found that lonely individuals hold atypical expectations of emotion transitions for both themselves and others. In addition, lonely individuals' mental models of others' emotion dynamics were less accurate, less variable (potentially indicating lowered confidence), and more volatile. Lonely individuals also exhibited an "anti-positivity" bias, viewing their own positive emotional states as especially fragile. They further showed constrained, and thus potentially less certain, estimates of the likelihoods that they themselves would move from one emotional state to another. Although the present data cannot decisively disentangle subjective confidence from scale-use bias, both interpretations converge on the idea that lonelier individuals may treat their social-transition judgments more cautiously, whether because they are uncertain or simply because they avoid committing to extremes. These patterns may contribute to difficulties fostering the sense of understanding and shared connection that typically anchors smooth social interactions and relationships.

Our data indicate that loneliness is linked to disruptions in emotion transitions for both the self and others. First, we found that lonely individuals exhibit atypical and idiosyncratic internal models, such that their

ratings of transitions between emotions were uniquely less similar from the norm compared to their non-lonely peers. This finding is in line with past research showing that lonely individuals tend to process social and emotional information in idiosyncratic ways[16,20]. For instance, lonely individuals show less typical neural representations and use idiosyncratic language when describing celebrities, suggesting that they might hold representations that deviate from cultural typicality[38]. Our findings corroborate and extend this prior work by suggesting that loneliness is also characterized by idiosyncratic emotion transition representations for both the self and others, further elucidating the cognitive atypicalities that characterize the lonely mind and hinder shared understanding that is essential for social connection.

Lonely individuals also showed reduced accuracy when estimating the emotion transitions of others. Replicating prior work, ratings provided by lonely participants were less aligned with a "ground truth" proxy of actual emotion transition likelihoods of others[12,13]. Such inaccuracies may have meaningful consequences, given that accurately anticipating emotion transitions of others has been associated with social well-being in both dyadic and communal relationships[12]. Inaccuracies in predicting others' emotion transitions can lead to stilted social interactions as they can result in miscalibrated behavior, such as steering a conversation in a direction that one's interaction partner may find inappropriate. Such repeated misattunements could lead to persistent interpersonal friction, reinforcing an already heightened sense of disconnection in lonely individuals.

Given the atypicality and inaccuracy nature of lonely individuals' predictions of others' emotion transitions, we examined whether these miscalibrations stem from a common strategy: anchoring on the self to predict others. This strategy is widely used[13–15] and can be rational and useful[32] but becomes less effective when an individual is atypical themselves[39]. We found that lonely individuals, in fact, anchored *less* on themselves when predicting others' emotion transitions, thus suggesting that their inaccuracies in predictions are not solely due to this strategy. These findings suggest two complementary possibilities. On the one hand, lonely individuals may have some awareness of their atypical emotion transitions and thus reduce reliance on their internal models when predicting others as an adaptive strategy. On the other hand, lonely individuals may have a reduced ability to simulate their own emotional experiences for use in

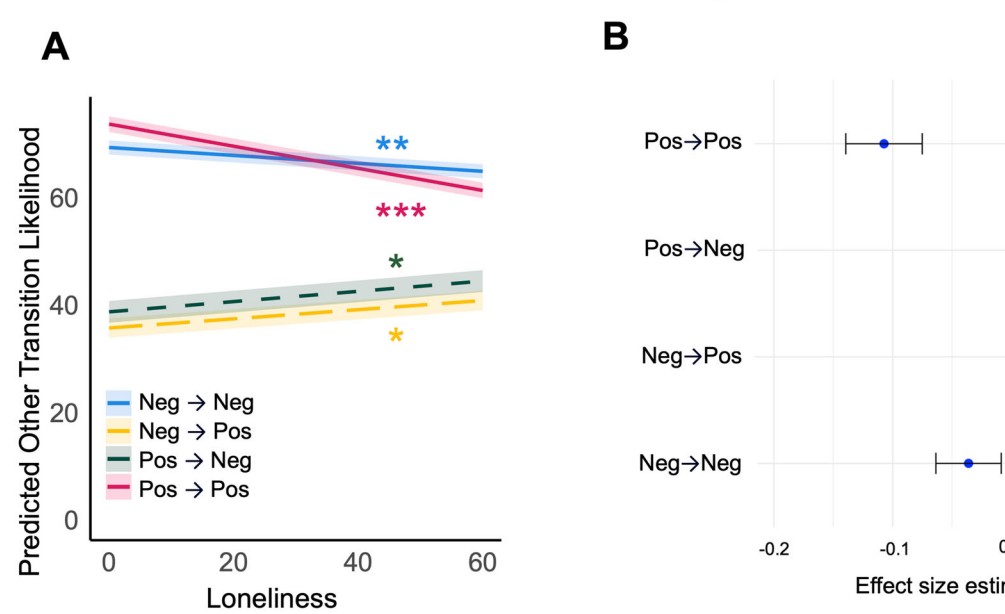

**Loneliness and likelihood of transition by emotion valence**
*Other ratings*

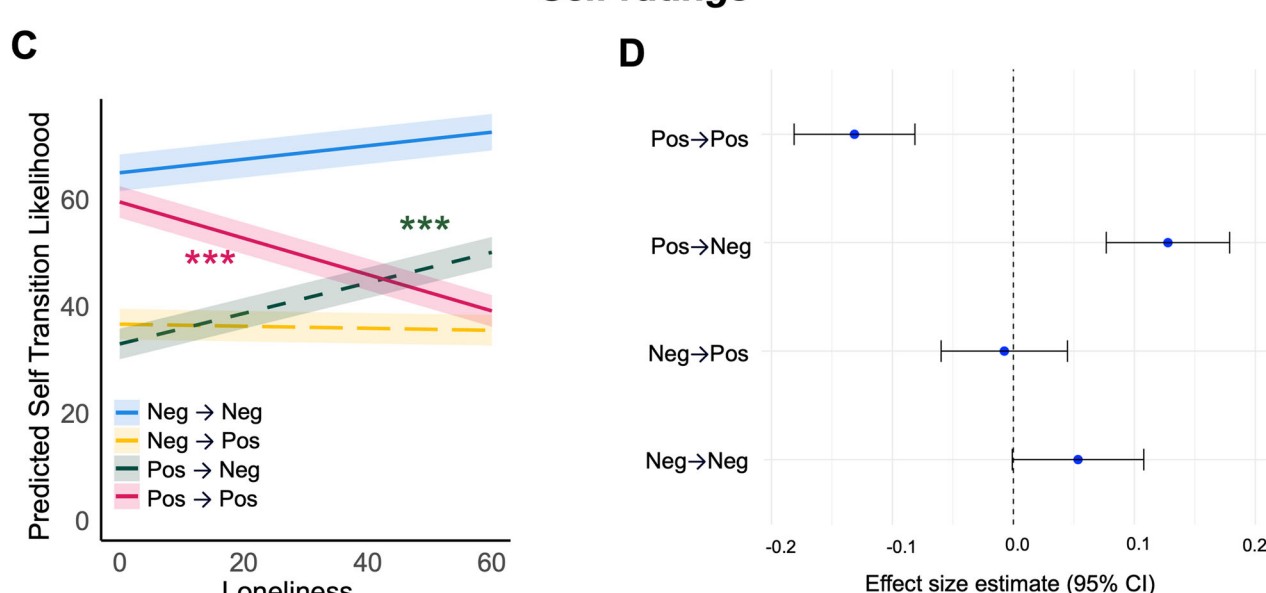

**Loneliness and likelihood of transition by emotion valence**
*Self ratings*

predicting others' emotions, perhaps related due to emotional numbness or alexithymia. These possibilities are in line with past work showing that lonely individuals display more differentiated neural representations of the self and others in brain regions implicated in social processing[20]. The potential awareness of atypicality may further reinforce feelings of isolation, as social connection hinges partially on perceiving the self's internal states as aligned with others' states[40,41]. Accordingly, lonely individuals' diminished sense of self-efficacy in social or emotional contexts[42] could discourage self-referential reasoning. Whether conscious or implicit, our findings suggest an adaptive but ultimately incomplete compensatory response: lonely

individuals may reduce self-anchoring because their emotional models differ from the norm but seem to not have an effective alternative basis for predicting others' emotional dynamics. Thus, even though loneliness is associated with less self-anchoring, lonely individuals' predictions remain inaccurate.

Why might lonely individuals' predictions for others (and themselves) be atypical, and why do their models for others remain inaccurate even when they are not anchoring on their own atypical models? One clue arises from our finding that lonely individuals are more likely to expect others to move between valence states and less likely to expect that they will remain in

**Fig. 4 | Loneliness is associated with perceptions that others are more volatile and with perceived fragility of positive states for the self. A** Lines represent the estimated simple slopes from a meta-analysis of mixed-effects models predicting emotion transition ratings from loneliness separately for each valence transition category. Higher loneliness is associated with greater expectations that others will transition between positive and negative states (positive to negative in dashed green line, negative to positive in dashed yellow line) and lower expectations that others will remain in the same valence state (solid lines; positive to positive in pink, negative to negative in blue). In other words, lonely individuals expect more volatility in others' emotion transitions. Shaded bands indicate 95% confidence intervals ($N = 6$ studies). Asterisks indicate significance levels of the slope differing from 0: $p < 0.05$ (*), $p < 0.001$ (***). **B** The meta-analytic fixed-effect estimates for the relationship between loneliness and emotion transition ratings, separately for each valence transition category, are visualized. Each point represents the estimated standardized beta coefficient, with horizontal error bars denoting the 95% confidence intervals

($N_{Study1} = 113$; $N_{Study2} = 185$; $N_{Study3} = 376$; $N_{Study4} = 91$; $N_{Study5} = 68$; $N_{Study6} = 41$). **C** Lines represent the estimated simple slopes from a meta-analysis of mixed-effects models predicting emotion transition ratings from loneliness separately for each valence transition category. Higher loneliness was associated with greater expectations that the self will transition into a negative state from a positive state (in green) and lower expectations that the self will transition into a positive state from a positive state (in pink). In other words, lonely individuals perceive positive states as less stable for themselves. Shaded bands indicate 95% confidence intervals ($N = 7$ studies). Asterisks indicate significance levels of the slope differing from 0: $p < 0.001$ (***). **D** The meta-analytic fixed-effect estimates for the relationship between loneliness and emotion transition ratings separately for each valence transition category are visualized. Each point represents the estimated standardized beta coefficient with horizontal error bars denoting the 95% confidence intervals ($N_{Study1} = 113$; $N_{Study2} = 185$; $N_{Study3} = 376$; $N_{Study4} = 91$; $N_{Study5} = 68$; $N_{Study6} = 41$; $N_{Study7} = 856$).

the same valence state. One possibility is that lonely individuals might perceive a conversation partner as shifting unpredictably between emotion states, such as from calm to irritable or irritable to cheerful. Another possibility is that they feel more uncertain about others' stability, leading them to interpret ambiguous signals as changes in valence. In either case, this perceived volatility could complicate social planning and reduce the sense of safety in interpersonal contexts. This may also partially explain why lonely people hold more negative expectations of others, particularly seeing them as less trustworthy, fair, and reciprocal[43], as volatility can undermine trust[44]. However, despite these negative expectations, lonely individuals do not necessarily disengage from social interactions. Instead, they may continue to value social engagement, even after negative feedback[45]. Further, while lonely individuals harbor negative expectations about social interactions and other people, they also express a positive inclination to connect with others[43]. Thus, loneliness can be thought of as a state in which the need for social connection and belonging is unmet[46], creating a heightened desire for social reconnection[47,48]. This paradox, wherein loneliness fosters both a wariness of others and a persistent longing for connection, may contribute to the instability of social predictions, reinforcing a cycle in which lonely individuals struggle to build and maintain meaningful social bonds.

Although lonely individuals viewed others as shifting between valences in volatile ways, they exhibited a different pattern for themselves. Specifically, loneliness was associated with a lower likelihood of remaining in a positive state and a higher likelihood of transitioning from a positive to a negative state. These patterns align with the documented link between loneliness and heightened sensitivity to negative affect[34], experience of negative emotion[19], and rumination strategies[49]. Our results suggest that this previously observed negativity bias in loneliness may not only stem from heightened sensitivity to negative affect but also from a diminished positivity bias and a reduced sense of stability when experiencing positive emotion states. This instability in maintaining positive emotions may further contribute to the emotional and social challenges that reinforce loneliness over time.

## Limitations
Although our multi-study, meta-analytic approach provides robust evidence of our findings, several limitations remain. First, the magnitude of the effects varied across individual samples, and in some cases the direction did as well. Although our meta-analytic approach accounts for such heterogeneity to yield a robust overall estimate, future research could explore study-level moderators more systematically. Furthermore, our data cannot determine causal direction. It is possible that prolonged loneliness causes disruptions in emotional beliefs. Alternatively, or simultaneously, having an atypical internal model of emotion transitions could hamper one's social life, ultimately leading to loneliness. Future studies utilizing longitudinal or experimental investigations can further elucidate whether interventions that reshape emotion transition expectations could decrease loneliness, or whether reductions in loneliness prompt more normative emotion transition beliefs. Although prior work with momentary assessment methods

corroborates that lonely individuals tend to be less accurate at predicting others' emotion transitions in everyday life[13], we do not have comparable data on lonely individuals themselves and thus cannot confirm the degree to which their self-forecasts may be (in)accurate. Future studies utilizing experience sampling methods could clarify whether lonely individuals' predicted shifts align with their actual emotion transition patterns as experienced daily. Testing whether lonely individuals genuinely experience fragile positive states could advance our understanding of whether these patterns reflect perceptual biases or genuine individual differences in emotion regulation and experience.

Furthermore, our findings suggest that lonely individuals show reduced accuracy in predicting the emotion transition likelihoods of *average* others. One possibility is that due to homophily[50], lonely individuals tend to interact more frequently with other lonely or otherwise "atypical" individuals[51], which could further distort their perception of a "typical other". If a lonely individual's primary social circle consists of people who, like them, are more atypical, their internal model of how an "average" person behaves might deviate from the broader population but still be an accurate representation of their social ties. In this sense, their elevated estimates of emotional volatility may not be wholly inaccurate but rather reflect the micro-environment in which they live. Future studies could explore the composition of lonely individuals' social networks, investigating whether relatively homogeneous groups of lonely people reinforce each other's atypical expectations, ultimately shaping beliefs and behaviors in a self-sustaining cycle. Additionally, the nature of our "other" target was a generic or an average other. Future work could examine how lonely individuals predict the emotion dynamics of close friends, romantic partners, family members, or newly encountered others. It remains possible that certain relationship contexts could mitigate or exacerbate lonely individuals' atypical expectations. Similarly, investigating the interplay between loneliness and specific interpersonal motives in different relational contexts, such as the desire to impress a superior or solicit comfort from a loved one could refine our understanding of how these atypical emotion-transition models shape real-world social interactions. Since lonely individuals report a heightened desire to support their networks[43], exploring how they anticipate loved ones' emotion states in more personal contexts could provide a richer picture of these processes.

Past research suggests significant comorbidity among loneliness, stress, anxiety, and depression[52,53], indicating that these variables may jointly influence individuals' emotion-transition predictions. Despite this co-occurrence, our findings on loneliness diverge from patterns that have been observed in depression, suggesting that loneliness may have unique effects on emotion prediction processes. For instance, prior work has shown that individuals with depression tend to be *more* rather than less accurate at tracking their loved ones' emotion transitions, though this is accompanied by a negativity bias such that they also rate others' emotions as more negative, and their tracking accuracy is especially sensitive to shifts toward the negative[54]. In contrast, our findings suggest that lonely individuals are *less* accurate and do not necessarily perceive others as more negative but

rather more emotionally volatile. This suggests the presence of cognitive mechanisms that may be specific to loneliness. While our investigation was guided by theories and empirical insights focused on loneliness, future research would benefit from examining these co-occurring psychological states concurrently to better elucidate their shared and distinct influences.

A growing body of literature suggests that loneliness covaries with demographic variables such as age, gender, and ethnicity[55–58], although individual findings are sometimes mixed[57]. Given that such characteristics also shape interpersonal goals and emotion-regulation habits (e.g., older adults prefer to attend to positive information[59], emotion norms are often gendered[60]), it is plausible that they may also influence how loneliness affects individuals' perceptions of their own and others' emotion transition likelihoods. Given that the current investigation was neither designed nor powered to test these effects, future work should model demographic variables simultaneously with loneliness to clarify their combined influence on emotion transition expectations.

From an applied perspective, our results raise the possibility that helping people recalibrate their models of emotion transitions could be one route to alleviating loneliness. Although it is currently unknown how malleable these models are, interventions could attempt to train lonely individuals to recognize more stable positive states in themselves or to see others' emotions as more predictable than they initially assume. Mindfulness-based[61] or cognitive-behavioral approaches[62] that help restructure beliefs around the persistence of positive affect and the realistic patterns of emotional change might yield more attuned social forecasting, and in turn, reduce the tendency to withdraw from or misread social partners.

## Conclusions

In sum, our findings illustrate the ways in which loneliness distorts emotional predictions for both the self and others. Lonely individuals perceive others in more atypical, inaccurate, and uncertain ways and expect greater volatility in other's emotion transitions. At the same time, they also see themselves as atypical and deviate from the common strategy of anchoring on one's own emotion transition patterns to predict others' emotion transitions. Additionally, they perceive their own positive emotional states as especially unstable. These distortions likely contribute to difficulties in navigating social interactions and relationships, further exacerbating social disconnection and undermining well-being.

## Data availability

Data from Studies 1-4 are publicly available at: https://osf.io/8mtxq/. Data from Study 5 are available at https://osf.io/7gjz9/ and data from Studies 6 and 7 are available for Studies 6 and 7: https://osf.io/7ybwr/.

## Code availability

All analysis code is available at https://osf.io/7gjz9/.

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

## Acknowledgements
We thank Jay Campanell, Zack Culver, Makyala Lui, and Preyashi Poddar for providing support with data collection for Study 5. We thank Elyssa Barrick for sharing her materials with us. This work was supported by a Brain and Behavior NARSAD Young Investigator Grant #32767 to E.C.B. The funders had no role in study design, data collection and analysis, decision to publish or preparation of the manuscript.

## Author contributions
E.C.B. designed Study 5 with support from B.G.B. S.M.B. designed Study 6 and 7 with support from L.F.F. and E.F. B.G.B., C.L., and J.C.Z. collected the data for Study 5. S.M.B., L.F.F., and E.F. collected the data for Study 6 and 7. A.Q.M.S. analyzed the data with support from E.C.B., M.E.S., S.M.B., and H.Y. A.Q.M.S., E.C.B., and S.M.B. wrote the manuscript with feedback from all authors. A.Q.M.S. made the figures with support from M.E.S.

## Competing interests
The authors declare no competing interests.
