## [Transparent Peer Review file · Communications Psychology]

Loneliness is associated with unstable and distorted emotion transition predictions

Corresponding Author: Ms Ava Ma de Sousa

Version 0:

Decision Letter:

Dear Ms Ma de Sousa,

Thank you for your patience during the peer-review process. Your manuscript titled "Loneliness is associated with unstable and distorted emotion transition predictions" has now been seen by 2 reviewers, and I include their comments at the end of this message. As you will see below, both reviewers find your work valuable but have raised some important points. Both reviewers' comments are clear, and I will not reiterate them here. We are interested in the possibility of publishing your study in Communications Psychology but would like to consider your responses to these concerns and assess a revised manuscript before we make a final decision regarding publication.

We therefore invite you to revise and resubmit your manuscript, along with a point-by-point response to the reviewers. Please highlight all changes in the manuscript text file.

Editorially, we consider it important that the revised manuscript conduct the additional analyses requested by the reviewers accompanied by the additional statistical reporting.

I am attaching an Editorial Requests Table that details critical reporting requirements for the revised manuscript. Please attend to each item and ensure your manuscript is fully compliant. If your revised manuscript is not aligned with these requests on major issues, such as those concerning statistics, it may be returned to you for further revisions without re-review.

Please submit the following items:

- Revised manuscript
- Point-by-point response to the referees' comments
- Cover letter (as a separate document)
- <https://www.nature.com/documents/nr-reporting-summary.zip>>Nature Research Reporting Summary
- <https://www.nature.com/documents/nr-editorial-policy-checklist.pdf>>Editorial Policy Checklist
- Completed Editorial Request Table (attached).

via this link: Link Redacted .

Additional guidance is available in our style and formatting guide Communications Psychology formatting guide.

Best regards,

Chuan-Peng

Hu Chuan-Peng, PhD
Editorial Board Member
Communications Psychology
orcid.org/0000-0002-7503-5131

REVIEWER EXPERTISE:

Reviewer #1 (social cognition, emotion)

Reviewer #2 (social cognition, meta-analysis)

REVIEWER REPORTS:

Reviewer #1 (Remarks to the Author):

Ma de Sousa et al. present a mega-analytic investigation of the relationship between loneliness and predictions of emotion transitions. Combining data across seven studies, they find that loneliness is associated with less typical, less accurate, more volatile, and less confident ratings of emotion transitions in the self and/or generic others.

This paper addresses a clearly articulated and interesting research question with a large amount of behavioral data from several different sources. The analyses are appropriate, and the conclusions seemed well-justified by the data. Below I note a few issues which it might be worth considering further:

1. The paper's operationalization of confidence in terms of the standard deviation of transition ratings is reasonable, but confidence is not the only reason these standard deviations might vary across people. For example, something as simple as scale use bias – e.g., a preference for selecting extreme responses – could also contribute variance to this measure. Without additional convergent evidence that the results are specifically confidence driven, the authors should consider tempering the conclusions made based on this operationalized variable.
2. The paper does a good job of examining multiple sources of variation in the emotion transitions. However, it might be helpful to offer more detail about the specific transitions which drive the results. For example, which transitions are lonely participants least accurate about in their ratings of others, or most atypical in their self-ratings? I realize that these results might be less generalizable, because the states vary across studies, but adding some descriptive heatmaps or network graphs to the supplement might enrich the paper for interested readers.
3. The study finds that lonely people are less typical in their emotion transitions. But due to the way the data were analyzed, it's hard to tell if all lonely people were atypical in the same way, or in different ways. The authors do identify one shared way in which they were atypical – i.e., that they viewed their emotions as less stably positive. However, the reported results are not sufficient to gauge how much of the typicality results are explained by this effect. I would suggest adding the following analysis to clarify this issue: Within each study, compute the pairwise similarity between participants' emotion transition ratings. Then use representational similarity analysis (RSA) with an Anna Karenina-style predictor (see Finn et al., 2020) based on loneliness scores. This should reveal if higher loneliness scores are associated with greater inter-participant variation in emotion transition predictions.

Reviewer #2 (Remarks to the Author):

In this study, the authors analyzed data from 7 different studies (4 publicly available, 3 new samples), testing the association between mental models of emotion transitions for self and others and loneliness using a meta-analytic approach (see comment below about mega-analysis terminology). They find that loneliness is associated with atypical emotion transitions expectations for self and other, less accurate emotion transition predictions, less self-anchoring when making emotion transition predictions for others, greater volatility in their emotion transition predictions, and less stability in positive states.

Overall, I'm very enthusiastic about this manuscript. It's fascinating, rigorous, and well-written. Not to mention, it sheds light on the socio-affective correlates of an experience that has strong ties to health and well-being. I have the following comments for the authors:

- My understanding is that "mega-analysis" refers to analyses that pool raw data across multiple studies or that conducts a meta-analysis over multiple individual meta-analyses. The analysis that appears to have been performed is a standard meta-analysis; that is, study-specific estimates were generated and these study specific estimates were submitted to meta-analysis (as opposed to individual subject data being pooled across all studies). Of course, if I'm mis-understanding the analytic approach, more detail in the methods would be appreciated.
- I'm curious about the Pomona sample who reports the second highest mean loneliness scores (almost 2x as high as some of the other samples) and appears to be a bit of an outlier in some of the associations tested (e.g., in study 1, they're the only sample to show a positive association between loneliness and typicality). Are there any reasons why this sample would be outlying? Related, did the authors formally test for outlying/influential studies across the analyses?
- Can the authors report some measure of meta-analytic heterogeneity (e.g., I²) with their estimates?
- Can the authors report alpha or omega for the loneliness scale across studies?
- Some of the language could better reflect what was actually measured in the task. For example, throughout, the authors refer to "likelihood of transition," when for the self-ratings, it's not the empirical/ground truth likelihood, but the expectation of transition.
- As the data are correlational, I'm curious about third variables that could potentially explain or be contributing to the associations. For example, loneliness changes as a function of age and perhaps emotion transition models too as people accumulate social experiences across different types of relationships. Does controlling for age and/or other demographic features account for any variance in the associations? I'd also be interested to hear the authors speculate about other third variables that could be contributing the associations in the discussion, but I leave this to the authors' discretion.
- The authors offer interesting explanations for the reduction of self-anchoring as a function of loneliness. I wonder too if this could be due to emotional numbness, alexithymia, or a related, implicit process that doesn't imply a conscious choice to not use the self as a model. Further, to the extent that lonely individuals are aware of the atypicality of their emotion experience, not using the self as a model could be an adaptive strategy.
- I was intrigued to hear about the potential for intervention. I'm curious though whether there's any data to suggest these mental models of emotion transitions (or mental state transitions more generally) are very malleable.
- There are a large number of figures included across the main text and supplement, which at times makes it difficult to focus on the key findings. It may strengthen the manuscript to consolidate figures where possible (e.g., by combining related results or moving more secondary analyses to the supplement) to better emphasize the primary results and improve readability.

If you experience problems in linking your ORCID, please contact the [Platform](http://platformsupport.nature.com/)

Support Helpdesk.

Version 1:

Decision Letter:

Dear Ms Ma de Sousa,

Your manuscript titled "Loneliness is associated with unstable and distorted emotion transition predictions" has now been seen by our reviewers, whose comments appear below. In light of their advice I am delighted to say that we are happy, in principle, to publish a suitably revised version in Communications Psychology.

We therefore invite you to revise your paper one last time to address the remaining concerns of our reviewers and a list of editorial requests. At the same time we ask that you edit your manuscript to comply with our format requirements and to maximise the accessibility and therefore the impact of your work.

EDITORIAL REQUESTS:

SUBMISSION INFORMATION:

OPEN ACCESS:

*** TRANSPARENT PEER REVIEW:** Communications Psychology uses a transparent peer review system. On author request, confidential information and data can be removed from the published reviewer reports and rebuttal letters prior to publication. If you are concerned about the release of confidential data, please let us know specifically what information you would like to have removed. Please note that we cannot incorporate redactions for any other reasons.

*** CODE AVAILABILITY:** All Communications Psychology manuscripts must include a section titled "Code Availability" at the end of the methods section. We require that the custom analysis code supporting your conclusions is made available in a publicly accessible repository at this stage; please choose a repository that generates a digital object identifier (DOI) for the code; the link to the repository and the DOI must be included in the Code Availability statement. Publication as Supplementary Information will not suffice.

*** DATA AVAILABILITY:**

All Communications Psychology manuscripts must include a section titled "Data Availability" at the end of the Methods section. More information on this policy, is available in the Editorial Requests Table and at <http://www.nature.com/authors/policies/data/data-availability-statements-data-citations.pdf>

Link Redacted

Best regards,

Jennifer Bellingtier

Jennifer Bellingtier, PhD
Senior Editor
Communications Psychology

Hu Chuan-Peng, PhD
Editorial Board Member
Communications Psychology
orcid.org/0000-0002-7503-5131

REVIEWER EXPERTISE:

Reviewer #1 social cognition, emotion

Reviewer #2 social cognition, meta-analysis

REVIEWERS' COMMENTS:

Reviewer #1 (Remarks to the Author):

The authors have satisfactorily addressed the issues I raised in my initial review. I think this paper is now ready to make a valuable contribution to the literature.

Reviewer #2 (Remarks to the Author):

I appreciate the authors detailed responses to my comments, additional analyses, and revisions to the manuscript. The addition of the Anna K models are especially nice. All of my comments have been thoroughly addressed and I remain very enthusiastic about the work. My congratulations to the authors.

Response Letter

Manuscript: "Loneliness is associated with unstable and distorted emotion transition predictions"
[COMMSPSYCHOL-25-0182-T]

Comments from Reviewer 1

R1.1 The paper's operationalization of confidence in terms of the standard deviation of transition ratings is reasonable, but confidence is not the only reason these standard deviations might vary across people. For example, something as simple as scale use bias – e.g., a preference for selecting extreme responses – could also contribute variance to this measure. Without additional convergent evidence that the results are specifically confidence driven, the authors should consider tempering the conclusions made based on this operationalized variable.

Thank you for this comment. We agree that standard deviation in transition ratings could reflect response factors such as reluctance to choose extreme options rather than confidence itself. We have therefore softened the language throughout the manuscript to acknowledge this alternative possibility.

Here are examples of some of the edits that we made (with bold typeface indicating changed text):

"While lonely participants relied less on their own emotion transition patterns when predicting others' emotions, **they also showed a response pattern that may reflect reduced confidence**, suggesting they use a less stable or altered strategy for predicting others."
(Abstract)

"**Although the present data cannot decisively disentangle subjective confidence from scale-use bias, both interpretations converge on the idea that lonelier individuals may treat their social- transition judgments more cautiously, whether because they are uncertain or simply because they avoid committing to extremes.**" (Discussion, pg. 16-17)

R1.2. The paper does a good job of examining multiple sources of variation in the emotion transitions. However, it might be helpful to offer more detail about the specific transitions which drive the results. For example, which transitions are lonely participants least accurate about in their ratings of others, or most atypical in their self-ratings? I realize that these results might be less generalizable, because the states vary across studies, but adding some descriptive heatmaps or network graphs to the supplement might enrich the paper for interested readers.

Thank you for suggesting these additional analyses. We agree that it is interesting to investigate which specific transitions might drive our effects. Because the emotion transitions varied across studies, we focused our follow-up analysis on Studies 1, 2, and 3, which all used the same set of transitions. Given the very exploratory and post-hoc nature of these analyses, we combined data from these three studies to maximize generalizability and power.

To explore which transitions lonely individuals were least **accurate** in predicting, we took the absolute difference between a participant's ratings for others and our "ground truth" proxy (i.e., group average transition ratings for self) and correlated this absolute difference measure with loneliness for each transition type. A heatmap of the correlations is presented in Supplementary Fig. 7, with positive correlations indicating that loneliness is associated with reduced accuracy (i.e. greater difference between a participant's ratings for other transitions and the group average for self transitions). We found significant (at $p < .05$, uncorrected) positive effects for several transition types, such that loneliness was associated with *less* accuracy in transition estimates from 1) anxious to happy ($r = 0.091$, $p = .018$), 2) calm to irritated ($r = 0.100$, $p = .009$), 3) happy to happy ($r = 0.108$, $p = .005$), 4) happy to sluggish ($r = 0.091$, $p = .018$), 5) irritated to happy ($r = 0.077$, $p = .045$), 6) sad to irritated ($r = 0.086$, $p = .026$), 7) sad to sluggish ($r = 0.086$, $p = .026$), and 8) sluggish to happy ($r = 0.095$, $p = .014$).

Similarly, to explore which transitions lonely individuals were the **most atypical**, we first took the absolute difference between participant's **self** transition ratings and the group average ratings of self transitions. We then correlated this absolute difference measure with loneliness for each emotion transition type. A heatmap of these correlations is presented in Supplementary Fig. 3A, with positive correlations indicating that loneliness is associated with greater atypicality for self ratings. We found significant positive effects such that lonely individuals deviated more from average self estimates in transitions from 1) calm to calm ($r = 0.087$, $p = .025$), 2) calm to happy ($r = 0.112$, $p = .004$), 3) full of thought to happy ($r = 0.113$, $p = .003$), 4) happy to happy ($r = 0.158$, $p < .001$), 5) happy to irritated ($r = 0.101$, $p = .009$), 7) happy to sad ($r = 0.129$, $p < .001$).

We then repeated this procedure for **other** ratings, calculating the absolute difference between participant's ratings for other transitions and the group average ratings of other transitions. We then correlated this absolute difference measure with loneliness for each emotion transition type. A heatmap of the correlations is presented in Supplementary Fig. 3B, with positive correlations indicating that loneliness is associated with greater atypicality for other ratings. We found significant effects such that lonely individuals deviated more from average other estimates in transitions from 1) anxious to happy ($r = .086$, $p = .025$), 2) calm to irritated ($r = .091$, $p = .019$), 3) happy to happy ($r = .088$, $p = .023$), 4) sad to anxious ($r = .079$, $p = .041$), 5) sad to irritated ($r = .086$, $p = .026$), 6) sad to sluggish ($r = .093$, $p = .015$), and 7) sluggish to happy ($r = .092$, $p = .017$).

Although we hesitate to make strong claims based on these exploratory results, one pattern that we observed is that transitions involving "happy" are highly represented to be particularly less accurate and more atypical in lonely individuals. These results corroborate our main findings suggesting that loneliness may be characterized by a bias in positive emotions. We now include visualizations of these results in the Supplementary Information for interested readers.

We also copy the visualizations below for your convenience.

Accuracy (Supplementary Fig. 7):

Typicality (Supplementary Fig. 3):

R1.3. The study finds that lonely people are less typical in their emotion transitions. But due to the way the data were analyzed, it's hard to tell if all lonely people were atypical in the same way, or in different ways. The authors do identify one shared way in which they were atypical – i.e., that they viewed their emotions as less stably positive. However, the reported results are not sufficient to gauge how much of the typicality results are explained by this effect. I would suggest adding the following analysis to clarify this issue: Within each study, compute the pairwise similarity between participants' emotion transition ratings. Then use representational

similarity analysis (RSA) with an Anna Karenina-style predictor (see Finn et al., 2020) based on loneliness scores. This should reveal if higher loneliness scores are associated with greater inter-participant variation in emotion transition predictions.

We appreciate the reviewer's suggestion to use an Anna Karenina approach to better understand whether lonely people are atypical in the same way or are altogether idiosyncratic. We found that, indeed, that the results support the Anna Karenina principle. Lonely individuals' emotion transition ratings not only deviated from the group's norm but also were idiosyncratic in that their ratings were different in their own unique way. We now include the following sentences in the Results section that discusses these results:

“We also conducted Intersubject Representational Similarity Analysis (IS-RSA) to test whether these results follow an “Anna Karenina” pattern²⁴, or the notion that non-lonely individuals are all alike, but every lonely individual holds expectations of emotion transitions in their own idiosyncratic way. Our results support this hypothesis, suggesting that lonely individuals deviate not only from group norms and from non-lonely individuals but also from one another. In other words, loneliness was linked to a greater degree of interpersonal variability in emotion transition expectations. See “Anna Karenina Typicality Analysis” in the Supplementary Information and Supplementary Fig. 4 for more details.”

We have also added the detailed results in the Supplementary Information (SI pg. 4).

“[W]e conducted inter-subject representational-similarity analysis (IS-RSA) to participants' estimates of emotion-transition likelihoods. For each study, we calculated the Pearson correlation between every pair of participants' rating vectors to obtain a participant-by-participant similarity matrix. We did this twice, once for self ratings and once for other ratings. Next, we constructed a predictor matrix that embodied the “Anna Karenina” (“AnnaK”) principle for loneliness. For each pair of participants, we averaged their two ranks based on their loneliness score, with higher averaged ranks representing pairs in which at least one member is highly lonely. Following the approach of Finn et al. (2020), representational similarity was assessed by calculating Spearman's correlation of the vectorized upper-triangular portions of the emotion transition ratings similarity matrix and the loneliness AnnaK matrices (excluding diagonals). A negative ρ means that the more loneliness a pair contains, the less alike their emotion-transition ratings are.

The resulting ρ coefficient captures the degree to which the pattern of (dis)similarity dictated by loneliness explains the pattern of (dis)similarity in emotion-transition space: thus, a negative value indicates that the more lonely the members of a dyad are, the more dissimilar they are. Finally, as in other analyses, we then conducted an internal meta-analysis to obtain an overall estimate of the association between loneliness and typicality of ratings for self and other, respectively.

Self ratings. Results indicate support for the Anna Karenina pattern: non-lonely individuals were similar to one another in their transition ratings for self, while lonely individuals were idiosyncratic ($\rho = -0.135$, 95, SE = 0.044, $p = .002$, 95% CI [-0.221 -0.048], $\tau^2 = 0.007$, $I^2 = 57.83\%$). In other words, lonely individuals were atypical in their ratings for

self transitions their own unique way, in ways that are different from not only non-lonely individuals but also from other lonely individuals.

Other ratings. Results again support the Anna Karenina pattern: non-lonely individuals were similar to one another in their transition ratings for self, whereas lonely individuals were idiosyncratic ($\rho = -0.163$, $SE = 0.033$, $p < 0.001$, 95% CI [-0.228, -0.097], $\tau^2 < 0.001$, $I^2 = 0.00\%$). Here again, lonely individuals were atypical in their ratings for others' transitions, in ways that were different from both non-lonely individuals and other lonely individuals."

Comments from Reviewer 2

R2.1. My understanding is that "mega-analysis" refers to analyses that pool raw data across multiple studies or that conducts a meta-analysis over multiple individual meta-analyses. The analysis that appears to have been performed is a standard meta-analysis; that is, study-specific estimates were generated and these study specific estimates were submitted to meta-analysis (as opposed to individual subject data being pooled across all studies). Of course, if I'm mis-understanding the analytic approach, more detail in the methods would be appreciated.

We thank the reviewer for highlighting the confusion around the term "mega-analysis." The reviewer is correct in that our approach did not pool individual- level data from all samples. Instead, we generated summary statistics within each sample and then combined the estimates. We were initially reluctant to label our work as a "meta- analysis" because we did not survey a large body of research as a meta-analysis label might suggest. With that said, we note that our procedure is sometimes described as a "mini" or "internal" meta- analysis (Goh et al. 2016). Accordingly, we have replaced "mega- analysis" with "internal meta- analysis" or "mini- meta-analysis" throughout the manuscript to accurately reflect our analyses.

R2.2. I'm curious about the Pomona sample who reports the second highest mean loneliness scores (almost 2x as high as some of the other samples) and appears to be a bit of an outlier in some of the associations tested (e.g., in study 1, they're the only sample to show a positive association between loneliness and typicality). Are there any reasons why this sample would be outlying? Related, did the authors formally test for outlying/influential studies across the analyses?

Thank you for this comment. While preparing the revised manuscript, we discovered a reverse-coding error that affected two of our samples. After correction, the mean loneliness score is $M = 19.27$, $SD = 10.54$ for Study 6 (Pomona undergraduates), and $M = 24.51$, $SD = 12.50$ for Study 7 (the Prolific sample). These values are more similar to the mean and SD for loneliness in our other samples, as can be seen in the table below.

Study	Mean Loneliness	SD Loneliness
-------	-----------------	---------------

Sample 1	19.4	13.76
Sample 2	23.23	13.53
Sample 3	19.62	9.88
Sample 4	23.01	9.83
Sample 5	16.97	11.45

Importantly, correcting this error did not significantly affect our results. Specifically, while the specific estimates changed (and were updated throughout the manuscript), none of the results of our statistical tests (e.g., statistical significance at $p = 0.05$) were affected. We thank the reviewer for pointing this issue out, which uncovered this error, and we have rechecked all of our scripts to make sure there are no additional errors.

However, as noted by the reviewer, it remains true that Study 6 is the only study in which there was a positive association between typicality and loneliness. This could be due to contextual factors such as Pomona College being a small liberal arts college that might foster a more close-knit community, potentially altering how loneliness relates to typicality. With that said, we are hesitant to overinterpret these results, given that Study 6 also had the smallest sample size ($N = 41$) with a large confidence interval for the correlation between typicality and loneliness. Therefore, the observed results could potentially be an artifact of sampling error.

While we did not run dedicated tests for outlying or influential studies, our approach was to estimate the average effect across diverse contexts and samples. Therefore, we chose a meta-analytic approach to weigh each study by its precision, rather than to diagnose individual studies. We have added a note about this issue of heterogeneity in the manuscript:

“Although our multi-study, **meta**-analytic approach provides robust evidence of our findings, several limitations remain. **First, the magnitude of the effects varied across individual samples, and in some cases the direction did as well. Although our meta-analytic approach accounts for such heterogeneity to yield a robust overall estimate, future research could explore study-level moderators more systematically.**”

R2.3. Can the authors report some measure of meta-analytic heterogeneity (e.g., I^2) with their estimates?

Thank you for this suggestion. We have added τ^2 and I^2 values in our estimates throughout the paper where appropriate.

R2.4. Can the authors report alpha or omega for the loneliness scale across studies?

Thank you for this suggestion. We have added Cronbach's alphas for loneliness across studies, which we include in Table 1 (pg. 29). Alpha values for all studies range between 0.916 and 0.947.

R2.5. Some of the language could better reflect what was actually measured in the task. For example, throughout, the authors refer to “likelihood of transition,” when for the self-ratings, it’s not the empirical/ground truth likelihood, but the expectation of transition.

Thank you for this suggestion. We agree that the original language was unclear and potentially misleading. In response, we have revised the language throughout the paper to clarify the prospective nature of the task and ratings. Specifically, whenever it was ambiguous that the ratings were ‘expected’ or ‘perceived’ likelihood ratings, we updated the language to clarify this. Here are some example revisions:

“Lonely individuals’ **likelihood of transitioning** between emotions...” → “Lonely individuals’ **expectation of transition likelihood** between emotions...” (pg. 6).

“we first calculated the mean ratings for emotion transition likelihoods...” → “we first calculated the mean ratings for **expected** emotion transition likelihoods...” (pg. 9).

“lonelier individuals exhibit less typical emotion transitions for themselves...” → “lonelier individuals exhibit less typical **expectations of** emotion transitions for themselves...” (pg. 10)

R2.6. As the data are correlational, I’m curious about third variables that could potentially explain or be contributing to the associations. For example, loneliness changes as a function of age and perhaps emotion transition models too as people accumulate social experiences across different types of relationships. Does controlling for age and/or other demographic features account for any variance in the associations? I’d also be interested to hear the authors speculate about other third variables that could be contributing the associations in the discussion, but I leave this to the authors’ discretion.

Thank you for this question. To test the possibility that age could be a potential confounder, we first calculated correlations to test whether there are relationships between age and loneliness in three of our studies (Study 1, Study 2, and Study 7) that have diverse age ranges. Given that the remaining studies included only young adults in their 20s, we did not calculate the correlations between age and loneliness in those studies.

Our results indicate that loneliness was negatively associated with age:

Study 1: $r = -0.220$, $p = 0.021$

Study 2: $r = -0.098$, $p = 0.181$

Study 7: $r = -0.086$, $p = 0.012$

Given that loneliness did vary with age, we fit additional models for our typicality, anchoring, and accuracy analyses with age included as a fixed effect covariate. We include these results in the Supplementary Information (Typicality: Supplementary Fig 2.; Anchoring: Supplementary Fig.11; Accuracy: Supplementary Fig. 7).

Overall, we found that our results remained similar when controlling for age, although it does appear that age accounts for some variance across our results:

- Relationship between typicality in self ratings and loneliness, controlling for age: $\beta = -0.121$, $SE = 0.036$, $p < .001$, 95 % CI $[-0.192, -0.050]$, $\tau^2 = 0.003$, $I^2 = 39.36\%$
- Relationship between typicality in other ratings and loneliness, controlling for age: $\beta = -0.126$, $SE = 0.050$, $p = .013$, 95 % CI $[-0.224, -0.027]$, $\tau^2 = 0.082$, $I^2 = 46.54\%$
- Relationship between loneliness and accuracy, controlling for age: $\beta = -0.011$, $SE = 0.009$, $p = .184$, 95 % CI $[-0.028, 0.005]$, $\tau^2 = 0.0003$, $I^2 = 78.3\%$
- Relationship between anchoring on the self to predict other's ratings and loneliness, controlling for age: $\beta = -0.029$, $SE = 0.009$, $p < .001$, 95 % CI $[-0.046, -0.012]$, $\tau^2 = 0.0003$, $I^2 = 80.6\%$

While we hesitate to overinterpret these results given the focus of our study, we agree that the role of third variables is important to consider in future investigations. To that end, we have added the following in our Discussion section:

“Past research suggests significant comorbidity among loneliness, stress, anxiety, and depression^{44,45}, indicating that these variables may jointly influence individuals' emotion-transition predictions. Despite this co-occurrence, our findings on loneliness diverge from patterns that have been observed in depression, suggesting that loneliness may have unique effects on emotional prediction processes. For instance, prior work has shown that individuals with depression tend to be *more* rather than less accurate at tracking their loved ones' emotion transitions, though this is accompanied by a negativity bias such that they also rate others' emotions as more negative, and their tracking accuracy is especially sensitive to shifts toward the negative⁴⁶. In contrast, our findings suggest that lonely individuals are less accurate and do not necessarily perceive others as more negative but rather more emotionally volatile. This suggests the presence of cognitive mechanisms that may be specific to loneliness. While our investigation was guided by theories and empirical insights focused on loneliness, future research would benefit from examining these co-occurring psychological states concurrently to better elucidate their shared and distinct influences.

A growing body of literature shows that loneliness covaries with demographic variables such as age, gender, and ethnicity⁴⁷⁻⁴⁹, although individual findings are sometimes mixed⁵⁰. Because those same characteristics shape interpersonal goals and emotion-regulation habits (e.g., older adults' preference to attend to positive information⁵¹, gendered emotion norms⁵², it is plausible that they may influence how loneliness affects individuals' perceptions of their own and others' emotion transition likelihoods. As our studies were neither designed nor powered to test these effects, future work should model demographic variables simultaneously with loneliness to clarify their combined influence on emotion transition expectations.” pg. 22-23

R2.7. The authors offer interesting explanations for the reduction of self-anchoring as a function of loneliness. I wonder too if this could be due to emotional numbness, alexithymia, or a related, implicit process that doesn't imply a conscious choice to not use the self as a model. Further, to the extent that lonely individuals are aware of the atypicality of their emotion experience, not using the self as a model could be an adaptive strategy.

We thank the reviewer for the insightful suggestion regarding emotional numbness, alexithymia, and implicit processes potentially underlying reduced self-anchoring among lonely individuals. While our current data do not directly assess emotional numbness or alexithymia, these are compelling avenues for future research that may clarify additional mechanisms contributing to distorted emotion transition expectations in loneliness.

We have edited the corresponding paragraph in the Discussion to pose these possibilities..

"These findings suggest **two complementary possibilities**. On the one hand, lonely individuals may have some awareness of their atypical emotional transitions **and thus reduce reliance on their internal models when predicting others as an adaptive strategy**. On the other hand, or may have a reduced ability to simulate their own emotional experiences for use in predicting others' emotions, **perhaps related due to emotional numbness of alexithymia**. [...] **Whether conscious or implicit, our findings suggest** an adaptive but ultimately incomplete compensatory response: lonely individuals may **reduce self-anchoring because their emotional models differ from the norm** but seem to not have an effective alternative basis for predicting others' emotional dynamics. Thus, even though loneliness is associated with less self-anchoring, their predictions remain inaccurate." pg. 18

R2.8. I was intrigued to hear about the potential for intervention. I'm curious though whether there's any data to suggest these mental models of emotion transitions (or mental state transitions more generally) are very malleable.

Thank you for raising this important point. To our knowledge, it is currently unknown to what degree the mental models of emotion transitions are malleable. We have modified our discussion to acknowledge this:

"From an applied perspective, our results raise the possibility that helping people recalibrate their models of emotion transitions could be one route to alleviating loneliness. **Although it is currently unknown how malleable these models are, interventions could attempt to train lonely individuals** to recognize more stable positive states in themselves or to see others' emotions as more predictable than they initially assume." pg. 23

R2.9. There are a large number of figures included across the main text and supplement, which at times makes it difficult to focus on the key findings. It may strengthen the manuscript to consolidate figures where possible (e.g., by combining related results or moving more

secondary analyses to the supplement) to better emphasize the primary results and improve readability.

Thank you for raising this point. We agree that there are a large number of figures which could make synthesizing the results more challenging. Thus, we have edited the manuscript to make our manuscript more readable and digestible. Specifically, we have reduced our number of figures in the main manuscript from 8 to 4 by: 1) moving the confidence results to the Supplementary Information, 2) combining the self and other typicality results into one figure (Figure 1), and 3) combining the self and other valence results into one figure (Figure 4).